# Optimizing radionuclide sequestration in anion nanotraps with record pertechnetate sorption

Qi Sun[1], Lin Zhu[2,3], Briana Aguila [1], Praveen K. Thallapally[4], Chao Xu[5], Jing Chen[5], Shuao Wang[2], David Rogers [1] & Shengqian Ma [1]

The elimination of specific contaminants from competitors poses a significant challenge. Rather than relying on a single direct interaction, the cooperation of multiple functionalities is an emerging strategy for adsorbents design to achieve the required affinity. Here, we describe that the interaction with the target species can be altered by modifying the local environment of the direct contact site, as demonstrated by manipulating the affinity of pyridinium-based anion nanotraps toward pertechnetate. Systematic control of the substituent effect allows the resulting anion nanotraps to combine multiple features, overcoming the long-term challenge of $TcO_4^-$ segregation under extreme conditions of super acidity and basicity, strong irradiation field, and high ionic strength. The top material exhibits the highest sorption capacity together with record-high extraction efficiencies after a single treatment from conditions relevant to the used nuclear fuel (Hanford tank wastes, 95%) and legacy nuclear wastes (Savannah River Sites, 80%) among materials reported thus far.

[1] Department of Chemistry, University of South Florida, 4202 E. Fowler Avenue, Tampa, FL 33620, USA. [2] State Key Laboratory of Radiation Medicine and Protection, School for Radiological and Interdisciplinary Sciences (RAD-X) and Collaborative Innovation Center of Radiation Medicine of Jiangsu Higher Education Institutions, Soochow University, 215123 Suzhou, China. [3] State Key Laboratory of Environmentally Friendly Energy Materials, Southwest University of Science and Technology, Mianyang, Sichuan 621010, China. [4] Physical and Computational Science Directorate, Pacific Northwest National Laboratory Richland, Richland, WA 99352, USA. [5] Collaborative Innovation Center of Advanced Nuclear Energy Technology, Institute of Nuclear and New Energy Technology, Tsinghua University, Beijing 100084, China. Correspondence and requests for materials should be addressed to S.W. (email: shuaowang@suda.edu.cn) or to S.M. (email: sqma@usf.edu)

Nuclear fission is a mature technology that can make a vital contribution toward global low-carbon energy needs[1–6]. However, opposition over the safety and security of nuclear facilities and materials still remains and needs to be addressed if new nuclear facilities are to gain widespread acceptance. Much of the controversy surrounding nuclear power stems from the early days of industrial development and the Cold War arms race, which has left a legacy of contaminated sites that must be cleaned up[7–9]. In addition, the development of nuclear technologies has led to many cases of accidental and intentional release, with the accumulation of significant levels of radionuclides in the environment[10–12]. One potentially troublesome radionuclide is technetium-99 ($^{99}$Tc), a considerable fraction of which occurs as pertechnetate, $TcO_4^-$. This species, with a long half-life of 213,000 years, is highly mobile in groundwater and biological systems, leading to possible problematic spread of radiotoxic material following containment breaches[13–16]. In addition, the volatile nature of some $^{99}$Tc compounds (e.g., $Tc_2O_7$), generated during high-temperature vitrification of the nuclear waste, makes $^{99}$Tc problematic in the off-gas system design for vitrification facilities[17]. To this end, $^{99}$Tc management is important both in terms of ensuring safe disposal of radioactive waste from power generation, minimizing the off-site migration of the contaminated groundwater, and assisting in cleanup efforts of legacy waste sites, yet it remains an unmet challenge[17,18].

To address this need, recent efforts have focused on developing ion-exchange sorbents given their ease of implementation and high $^{99}$Tc recovery rate[19–24]. While elegant examples of materials embodying this strategy are now known, the battle to ameliorate $^{99}$Tc contamination is far from finished. Current industrial extraction technologies suffer from relatively low affinity and kinetics, as well as limited capacity for $TcO_4^-$. Given this, proper remediation of the legacy tank wastes is still pending. These facts underscore the need for material innovation capable of extracting radionuclides from complex media for environmental remediation and in order to close the nuclear fuel cycle.

Nature has been subject to millions of years of evolutionary experimentation to recognize specific metal ions with high sensitivity, whereby multiple functionalities are assisted with the direct binding sites. Chemical efforts to mimic such partnership have led to numerous advances in control of binding affinity. As a sophisticated example, supramolecular techniques have been utilized to engineer cations with intriguing properties that display particular innovation and hold promise as potential game-changing technologies in the field of target ion extraction. The selectivity emerges from the concerted efforts of the binding partners[25–27]. Thus, it is well conceivable that some structural modifications to a site separate from the place of direct contact of the association partners may well affect their binding strength, thereby providing the needed ability to recognize target anions under highly competitive conditions. While success was encountered using these receptors in the context of a classic liquid–liquid extraction approach, to meet the challenges posed by the immense volumes of wastewater, sorbent materials would allow for an increase in operational ability[28–34]. With these in mind, we envisioned that if distinct functionalities can be incorporated into a highly porous cationic framework, this may enable a new level of control over ion pair recognition and extraction efficiency within a sorbent material with a high density of readily accessible ion-exchange sites. This is also anticipated to afford new anion receptors with fast kinetics together with high uptakes, which surmount the prime issues of low efficiency encountered by commercial resins as a result of low surface area and low density of accessible ion-exchange groups.

To explore the validity of this hypothesis, we were motivated by the exploration of porous organic polymers (POPs) for this ambitious task. This class of porous material has moved to the forefront of materials research due to their modular nature. POPs exhibit tailorable porosity and tunable functionality, and are well-suited for a wide range of applications including gas storage/separation, catalysis, hazard contaminants removal, and sensing, to name a few[35–41]. The potential to deploy POPs as solid extractors is related to the ease of tunability with regard to their composition[42–44]. In this contribution, we chose pyridinium as the cation building unit to demonstrate the proof-of-concept, given its chemical stability, amenable synthesis, together with easy-to-get properties[45]. To optimize the anion exchange performance and to establish the structure-property relationships, different anchor groups were de novo introduced (Fig. 1). In addition, to improve the uptake capacity and efficiency of the resultant anion nanotraps, porous sorbent materials were constructed by the functional pyridinium moieties alone, to conserve a sufficiently high density of accessible exchange sites. Therefore, the resultant materials combine the advantages of high uptake capacity, selectivity, and removal rate of $TcO_4^-/ReO_4^-$ ions, making them promising for decontamination of polluted water and remediation of nuclear waste. Through detailed studies, we showed that the adsorbent constructed with a dimethylamino group in the para position relative to pyridinium displayed extraordinary affinity for $TcO_4^-$ with a record-breaking uptake capacity and selectivity under extreme conditions of super acidity and basicity, as well as high concentrations of competitors. These insights are meant to guide further research, which should be devoted to increasing the binding affinity of adsorbents toward target ions.

## Results

**Material synthesis.** To target the desired materials, our initial step was to construct pyridine moieties with various anchor groups into porous frameworks, due to the low solubility of pyridinium in organic solvents, which is detrimental to yield highly porous polymers. To construct these moieties into porous polymers, we decided to use free radical induced polymerization with the following considerations: (i) the resultant highly cross-linked polymer chains and thus the rigidity of the matrix, reduce the ability of the pyridinium to congregate in hydrated domains, which decrease the affinity for more hydrated anions and thereby enhance the selectivity for less hydrated anions such as $TcO_4-$ [46]; and (ii) the adapted synthesis allows for the introduction of hierarchical porosity in the resultant polymers, which is expected to enhance the kinetics, thereby offering an opportunity to meet the challenges posed by the vast volumes of wastewater[47,48]. To this end, we further functionalized the pyridine moieties with vinyl groups, affording V-Py, V-$p$NH$_2$Py, and V-$p$N(Me)$_2$Py, respectively (Table 1). Under solvothermal conditions in dimethylformamide (DMF) at 100 °C, polymerization of these monomers in the presence of azobisisobutyronitrile (AIBN) gave rise to the porous pyridine-based polymers in quantitative yields, which were denoted as POP-Py, POP-$p$NH$_2$Py, and POP-$p$N(Me)$_2$Py, respectively. To introduce ion-exchange sites, the pyridine moieties in the resultant materials underwent a quaternization reaction with methyl iodide (CH$_3$I), yielding the porous ionic organic polymer of polymerized quaternary ammonium (PQA) salt PQA-Py-I, PQA-$p$NH$_2$Py-I, and PQA-$p$N(Me)$_2$Py-I, respectively.

Among the synthesized porous materials, POP-$p$N(Me)$_2$Py and corresponding quaternization product PQA-$p$N(Me)$_2$Py-I were chosen as representative samples for thorough illustration. As probed by scanning electron microscopy (SEM) and transmission electron microscopy (TEM), it is revealed that no noticeable morphological changes occurred after quaternization, both

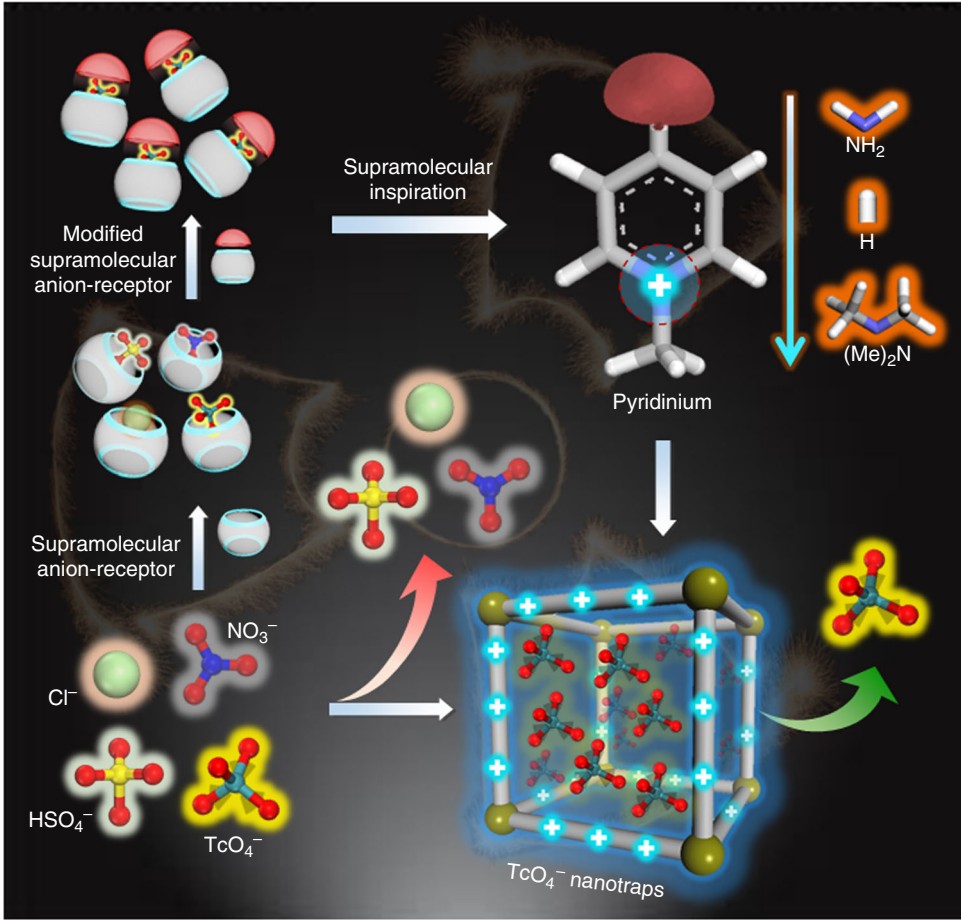

**Fig. 1** Anion nanotraps for $TcO_4^-$ removal. Illustration of optimizing pyridinium based anion nanotraps for $TcO_4^-$ recognition inspired by supramolecular technology

**Table 1 Structure of building units and textural parameters of various pyridinium functionalized hierarchical porous polymers**

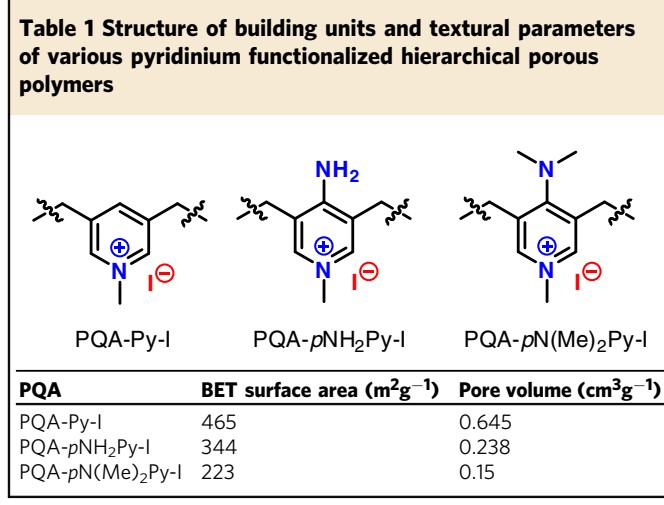

| PQA | BET surface area ($m^2g^{-1}$) | Pore volume ($cm^3g^{-1}$) |
|---|---|---|
| PQA-Py-I | 465 | 0.645 |
| PQA-$p$NH$_2$Py-I | 344 | 0.238 |
| PQA-$p$N(Me)$_2$Py-I | 223 | 0.15 |

POP-$p$N(Me)$_2$Py and PQA-$p$N(Me)$_2$Py-I feature aggregates comprising interconnected nanoparticles with sizes of 50–100 nm (Supplementary Fig. 1 and 2). Structural details of these materials were elaborated by solid-state $^{13}C$ NMR spectroscopy. As seen in Supplementary Fig. 3, the characteristic vinyl $^{13}C$ resonance located in the range of 100.0–110.0 ppm in the monomers disappears with the concomitant appearance of the polymerized vinyl group peak at 31.8 ppm, confirming the conversion of the monomer into the respective polymer[48]. In addition, the increased relative intensity of the peak at 44.9 ppm, ascribed to the methyl group from iodomethane ($CH_3I$), verified the occurrence of the quaternization reaction between the pyridine moiety on POP-$p$N(Me)$_2$Py and $CH_3I$. To provide additional proof, we employed X-ray photoelectron spectroscopy (XPS). The appearance of strong signals of iodine species at 629.3 and 617.8 eV for $I3d_{5/2}$ and $I3d_{7/2}$, respectively, confirmed the success of this transformation (Supplementary Fig. 4). To quantify the degree of quaternization, we evaluated the content of iodine species in PQA-$p$N(Me)$_2$Py-I by elemental analysis. The results showed that the weight percentage of iodine species in PQA-$p$N(Me)$_2$Py-I was 38.3 wt% (Supplementary Table 1), which means about 95% of the pyridine moieties were involved in the quaternization reaction, revealing the high throughput of this reaction. The details of the pore properties of the resultant materials were investigated by $N_2$ sorption isotherms collected at 77 K. It is shown that both samples exhibit type-I plus type-IV sorption curves, suggesting the retention of the pore structure (Supplementary Fig. 5). A steep step in the curve at relative pressure ($P/P_0$) less than 0.01 is due to the filling of micropores, whereas a hysteresis loop at $P/P_0$ in the range of 0.45~0.95 is mainly from the contribution of the sample mesoporosity, which was also verified by the pore size distribution[49]. The calculated specific surface areas were 535 and 223 $m^2g^{-1}$ for POP-$p$N(Me)$_2$Py and PQA-$p$N(Me)$_2$Py-I, respectively. The decreased surface area can be reasonably attributed to the increased mass after the functionality addition. The details of the characterization

of POP-Py and POP-$p$N(Me)$_2$Py, as well as their corresponding quaternization products are provided in the Supplementary Information (Supplementary Figs. 6–13).

**Sorption isotherms**. The full availability of ion exchange sites within the pores of these materials is indicated by a complete anion-exchange process between I$^-$ and Cl$^-$, which was verified by TEM-EDS mapping and XPS analyses (Supplementary Figs. 14–17). This feature spurred their exploration for radio-nuclide polluted water treatment. Sorption experiments were first performed to evaluate the working capacities of these materials. Due to the limited availability and the rarity of the purified isotope $^{99}$Tc, combined with its highly radioactive nature, per-rhenate (ReO$_4^-$) was first used as a nonradioactive surrogate for pertechnetate ($^{99}$TcO$_4^-$) given their similarities in both magnitude and trend between the solubilities[50]. The amounts of rhenium species enriched by these materials as a function of its concentration in the supernatant at the equilibrium state were determined by varying the initial concentrations from 25 to 800 ppm with a material-to-solution ratio of 0.5 mgmL$^{-1}$. An overnight reaction time was used to ensure that the ion exchange reached equilibrium. Results indicate that the equilibrium adsorption data of ReO$_4^-$ appeared to follow the Langmuir iso-therm model (Fig. 2a); sorption increases rapidly at low ReO$_4^-$ concentrations and slows appreciably as the sorption capacity of the adsorbent is approached, yielding high correlation coefficients ($R^2 > 0.99$). Exchange capacities up to 936, 806, and 1127 mg of ReO$_4^-$ per gram of polymer were observed for PQA-Py-Cl, PQA-$p$NH$_2$Py-Cl, and PQA-$p$N(Me)$_2$Py-Cl, respectively, almost 20 times that observed for the unmodified counterparts (POP-Py, POP-$p$NH$_2$Py, and POP-$p$N(Me)$_2$Py), affording 53, 46, and 37 mgg$^{-1}$, respectively. Given the greatly enhanced performances after quaternization, we establish that the high density of the ion exchange sites on the polymer backbone is the fundamental contributor to their high working capacities and the mechanism for uptake of ReO$_4^-$ is thus believed to be predominantly ion exchange.

The high extraction capacities for these materials prompted us to evaluate their ion exchange kinetics, which were determined from distilled water samples spiked with 50 ppm of rhenium in the form of ReO$_4^-$. Remarkably, when the solutions were treated with these materials at a phase ratio of 25000 mLg$^{-1}$, extremely rapid ion exchange and high uptake were recognized. All three materials reached their equilibrium exchange capacities within 60 min and in particular, 95.3% removal was achieved after 5 min for PQA-$p$N(Me)$_2$Py-Cl (Fig. 2b). This is in stark contrast to the lengthy contact times required for many sorbent materials, which

routinely range from several hours to as much as several days[51]. The fast kinetics should be an advantage for the continuous treatment of relatively large volumes of slightly contaminated water. We surmise that the impressive uptake capacity along with the fast removal rate is due to the high density of ion-exchange sites and intrinsic porosities allowing for rapid diffusion of ReO$_4^-$ throughout the materials. Notably, in addition to rapid saturation, PQA-$p$N(Me)$_2$Py-Cl also possessed a higher capacity, yielding an equilibrium value of 997 mgg$^{-1}$, in comparison with 849 and 642 mgg$^{-1}$ afforded by PQA-Py-Cl and PQA-$p$NH$_2$Py-Cl, respectively, positioning PQA-$p$N(Me)$_2$Py-Cl as the material with the highest ReO$_4^-$ uptake among all adsorbents reported so far (Fig. 2b and Supplementary Table 2). Intriguingly, the increase in gravimetric capacity in going from PQA-$p$N(Me)$_2$Py-Cl to PQA-Py-Cl and PQA-$p$NH$_2$Py-Cl is opposite to what would be expected from the density of ion-exchange sites. For instance, on the basis of elemental analysis, it was found to have a Cl$^-$ species content of 10.3, 11.1, and 12.5 wt%, for PQA-$p$N(Me)$_2$Py-Cl, PQA-Py-Cl, and PQA-$p$NH$_2$Py-Cl, respectively (Supplementary Table 3). We therefore attribute this anomalous increase in ReO$_4^-$ uptake capacity for PQA-$p$N(Me)$_2$Py-Cl to the higher electrostatic ion-pairing attraction energies between the cationic polymer backbone of PQA-$p$N(Me)$_2$Py-Cl and ReO$_4^-$ ions. To rationalize this assumption, we measured the distribution coefficient values ($K_d$, for definition, see the Experimental Section) of these materials toward ReO$_4^-$. Under the conditions of 10 ppm rhenium with a V/m of 1000 mL g$^{-1}$, the $K_d$ values were calculated and found to equal $1.13 \times 10^6$ (8.82 ppb), $5.44 \times 10^5$ (18.35 ppb), and $1.00 \times 10^7$ (1.00 ppb) for PQA-Py-Cl, PQA-$p$NH$_2$Py-Cl, and PQA-$p$N(Me)$_2$Py-Cl, respectively (number given in parentheses is the residual rhenium concentration), with PQA-$p$N(Me)$_2$Py-Cl showing almost more than an order of magnitude improvement in performance over the other two, indicative of its stronger affinity toward ReO$_4^-$ and thereby facilitating the ion exchange between Cl$^-$ and ReO$_4^-$.

Next, we assessed selectivity over common anions. The material's ability to competitively maintain efficiencies and rapid extraction rates at sub ppm levels, more typical of exposure cases, is one of the most important factors when evaluating a material for water treatment applications. Given the high environmental mobility, any accidental leaching of TcO$_4^-$ poses a real threat to groundwater contamination and aquatic life forms. As such, to access the ion-exchange properties associated with these materials, treatability studies were carried out using groundwater spiked with 1000 ppb of rhenium to simulate the contaminated water from on-site and off-site wells. PQA-$p$N(Me)$_2$Py-Cl was shown to remove >99.8% of ReO$_4^-$ from the groundwater, bringing rhenium concentrations to 1.2 ppb, implying that the

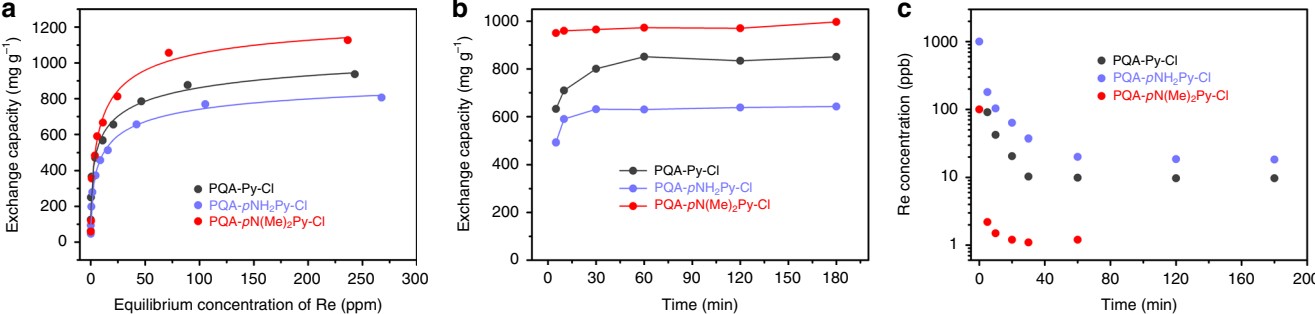

**Fig. 2** ReO$_4^-$ adsorption isotherms and kinetics investigations. **a** ReO$_4^-$ sorption isotherms for various adsorbents. The lines are fit with the Langmuir model; all the fits have $R^2$ values higher than 0.99. **b** The kinetics of ReO$_4^-$ adsorption from aqueous solution with an initial concentration of 50 ppm, at a phase ratio (V/m) of 25000 mLg$^{-1}$. **c** ReO$_4^-$ removal kinetics with an initial concentration of 1000 ppb at a V/m ratio of 5000 mLg$^{-1}$. Source data are provided as a Source Data file

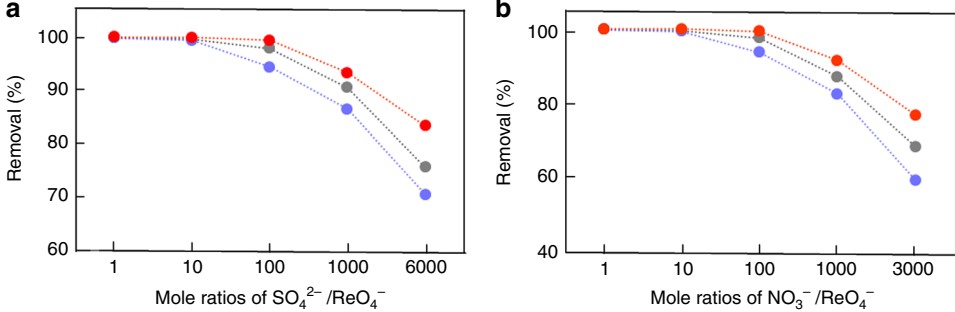

**Fig. 3** Selectivity evaluation. Effect of competing (**a**) $SO_4^{2-}$ and (**b**) $NO_3^-$ anions on the anion-exchange of $ReO_4^-$ by (gray) PQA-Py-Cl, (blue) PQA-$p$NH$_2$Py-Cl, and (red) PQA-$p$N(Me)$_2$Py-Cl (lines are guidelines for the eyes). Source data are provided as a Source Data file

interferences commonly found in surface waters have a minimal effect on its performance. It is worth mentioning that the residual rhenium concentrations treated by PQA-$p$N(Me)$_2$Py-Cl are 8 and 15 times lower than that of PQA-Py-Cl and PQA-$p$NH$_2$Py-Cl, respectively (Fig. 2c). In addition to the impressive removal efficiency, it is outstanding from the viewpoint of kinetic efficiency, reaching the equilibrium exchange capacity within 20 min and over 99.7% of the rhenium species removed within 5 min These results highlight the vast potential of PQA-$p$N(Me)$_2$Py-Cl in accomplishing radionuclide removal from groundwater.

Encouraged by the results above, we then evaluated wastewater samples. Compared to surface water samples, wastewater is often comprised of fewer competing ions but significantly higher concentrations. Given this, to evaluate the sorption selectivity of $ReO_4^-$ against the major anions $NO_3^-$ or $SO_4^{2-}$, we performed binary exchange reactions by varying the equivalent ionic fractions of these anions in reactant solutions. A plot of the removal efficiencies of $ReO_4^-$ against mole fractions of $NO_3^-$ or $SO_4^{2-}$ to $ReO_4^-$ clearly indicated that PQA-$p$N(Me)$_2$Py-Cl showed a remarkably high tolerance to $ReO_4^-$ sorption ability in the presence of $SO_4^{2-}$ and $NO_3^-$ competition with very high removal efficiencies and excellent $K_d$(Re) values (Fig. 3). Specifically, at $SO_4^{2-}$/$NO_3^-$: $ReO_4^-$ molar ratios of 1 to 100, >99.9% of $ReO_4^-$ ions were sequestered with $K_d$(Re) values in the range of $1.5 \times 10^5$–$6.6 \times 10^6$ mLg$^{-1}$. It is interesting that even with a tremendous excess of Na$_2$SO$_4$ ($SO_4^{2-}$: $ReO_4^-$ molar ratio at 6000), PQA-$p$N(Me)$_2$Py-Cl still retained a very good $ReO_4^-$ removal efficiency (83.4%) and high $K_d$(Re) value (5034 mL g$^{-1}$), indicative of an exceptional selectivity of PQA-$p$N(Me)$_2$Py-Cl for $ReO_4^-$ against $SO_4^{2-}$. This feature makes it an extremely attractive candidate to selectively remove $ReO_4^-$/ $TcO_4^-$ from waste solutions with high ionic strengths. By contrast, under identical conditions, PQA-Py-Cl and PQA-$p$NH$_2$Py-Cl afforded $ReO_4^-$ removal efficiencies of 75.7% and 70.6%, with corresponding $K_d$(Re) values of 3123 and 2402 mLg$^{-1}$, respectively, inferior to those for PQA-$p$N(Me)$_2$Py-Cl (Supplementary Fig. 19).

Under all conditions tested, the preference of these pyridinium-based sorbents towards $ReO_4^-$ complexation follows the same order: PQA-$p$N(Me)$_2$Py-Cl > PQA-Py-Cl > PQA-$p$NH$_2$Py-Cl. This trend may not be pinpointed to a single property change of the adsorbents but rather as a consequence of a complex interplay of several factors. For example, an increase in surface area may play a key role in the uptake capacity by providing a greater number of exposed exchangeable sites. However, in the present case, only a very weak correlation with the surface area of these materials could be established, thus suggesting that surface area is not the central factor determining the sorption performance; rather, the interaction strength between binding sites and guest species is a more important factor for determining guest uptake. To understand the lack of

dependence on the materials' textural parameters, we investigated the distribution of $ReO_4^-$ in solid samples. Elemental distribution mapping was performed by SEM. We found homogeneously and densely distributed rhenium species throughout each sample (Supplementary Fig. 20). To examine the binding behavior of $ReO_4^-$ in these adsorbents, XPS and IR spectroscopy were carried out. Comparison of Re4$f$ signals in the reacted samples (Re@PQA-Py-Cl, Re@PQA-$p$NH$_2$Py-Cl, and Re@PQA-$p$N (Me)$_2$Py-Cl), Re@PQA-$p$NH$_2$Py-Cl and Re@PQA-$p$N(Me)$_2$Py-Cl displayed very similar binding energies, giving Re4$f_{7/2}$ signals at 45.1 and 45.0 eV, respectively (Supplementary Fig. 21), which are lower than those in Re@PQA-Py-Cl (45.6 eV) and KReO$_4$ (46.1 eV). This presumably arises from the electron donating groups present in PQA-$p$NH$_2$Py-Cl and PQA-$p$N(Me)$_2$Py-Cl, which weakens the partial positive charge on the cations and thereby that on the rhenium species. These observations were further corroborated by the IR spectra (Supplementary Fig. 22), a slight red-shift of the $ReO_4^-$ antisymmetric vibration was detected in Re@PQA-$p$NH$_2$Py-Cl and Re@PQA-$p$N(Me)$_2$Py-Cl (893.8 cm$^{-1}$) compared to that in Re@PQA-Py-Cl (901.6 cm$^{-1}$) and KReO$_4$ samples (906.8 cm$^{-1}$). Regarding electronic factors, the introduction of electron donating groups facilitates the formation of kinetically labile complexes, which allows for rapid guest exchange. However, the relevant explanations cannot correlate with the observed trend, wherein the introduction of the amino group gave adverse results, with the lowest performance of the three materials. To explain this, we reasoned that, different from the dimethylamino group, the amino group acts both as a hydrogen bond donor and a hydrogen bond acceptor. Anions, such as Cl$^-$, are inclined to interact with it via hydrogen bonding, thereby compromising the ion-exchange capability for ReO$_4$ ions. Such an interaction between the amino group and the Cl$^-$ ion is expected considering the successful Cl$^-$ extraction solely on an amine-based neutral anion receptor[27].

**Density functional theory calculation studies**. We carried out quantum density functional theory (DFT) computations to understand the chemical basis of the binding selectivities toward $ReO_4^-$/$TcO_4^-$. Calculations on the truncated fragments shown in Fig. 4 were performed using M06 exchange and correlation functions with fine grid spacing as implemented in NWChem. A 6-311++G(2d,2p) basis was used for all atoms except technetium, which was treated with the def2-tzvp basis and associated empirical core potential. This combination of functional and basis set was shown to reproduce experimental geometries of $TcO_4^-$[52,53], providing more confidence in the results than could have been obtained with calculations on $ReO_4^-$. Full details of the quantum mechanical calculation are provided in the Supplementary Information.

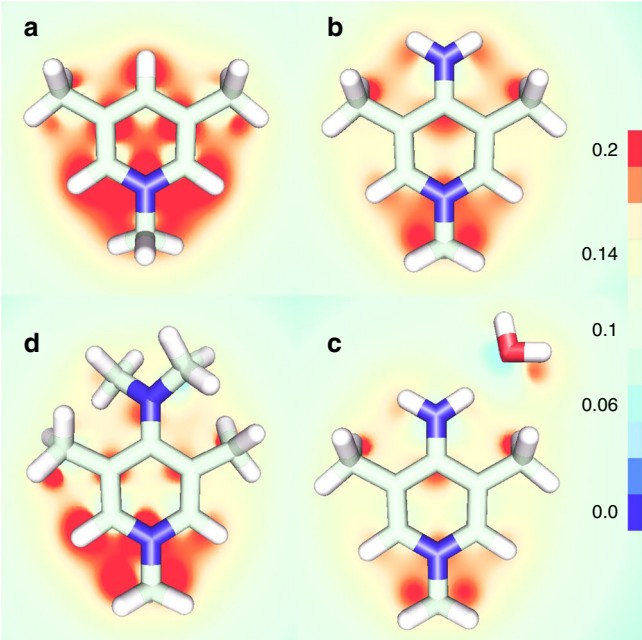

**Fig. 4** Electrostatic potential surface. Distribution of the charge density around the aromatic ring is shown by a plot of the electrostatic potential surface just below the molecule. The most favorable binding geometry for a triangular face of an anion has opposite orientations between the cationic building blocks of **a** PQA-Py-Cl, **b**, **c** PQA-$p$NH$_2$Py-Cl, and **d** PQA-$p$N(Me)$_2$Py-Cl. Methyl groups have been introduced to take the electrostatic effects of the polymer chain into account. Comparison of b and c shows some negative charge is donated to the ring system during formation of a hydrogen bond

In all molecules, the pyridinium nitrogen forms a center of positive potential. Concentration of the positive charges at meta-positions would suggest that the triangular faces of NO$_3^-$, HSO$_4^-$, and TcO$_4^-$ should orient directly overtop (Supplementary Fig. 23). Instead, the minimum energy geometries of a show that two of the anion oxygens seek to overlap with the lobes of positive potential on either side of the ring nitrogen closer to its attached methyl. These differences in geometry are significant, since all geometry optimizations started from anion positions containing an oxygen pointing toward the ring nitrogen. For d, the tertiary amine also has a large positive partial charge, and pulls anion oxygens toward itself. The resulting competition between these sites results in the smaller nitrate and sulfate anions stretching one oxygen toward each site, and not effectively using their third binding oxygen. In contrast, the longer bond length of TcO$_4^-$ allows it to simultaneously occupy positions next to the dimethylamino group and close to the ring nitrogen. This explains the opposite binding orientation to what would be expected based on placing oxygens over both meta- sites. b is interesting and in its place, we see a strong positive charge at the para- location with nearly zero charges meta- to the ring nitrogen. Structure c was included to understand the effect of electron donation to the ring system due to a hydrogen bonding of water to the side-chain amine. It can be seen that the overall potential of the binding site decreases slightly, which will result in less favorable anion binding.

Binding free energies were computed for Cl$^-$, TcO$_4^-$, NO$_3^-$, and HSO$_4^-$ to all four structures in Fig. 4 using the quasi-chemical method (see details in Computational Methods Section and Supplementary Table 4 and Fig. 23). The anion HSO$_4^-$ was used in place of SO$_4^{2-}$ because the computational results are

more reliable when comparing ions of equal valence. The difference in binding free energies between HSO$_4^-$ and SO$_4^{2-}$ is the same as the difference between the pKas of HSO$_4^-$ in bulk solution compared to HSO$_4^-$ bound to the polymer.

A large portion of the binding free energies is due to solvation effects. These increase polymer binding affinity of the larger (less hydrophilic) TcO$_4^-$ over the smaller Cl$^-$ and NO$_3^-$. Although our solvation free energy calculations were carried out for liquid water, the polymer environment is expected to be slightly more hydrophobic. This will give TcO$_4^-$ and additional boost relative to other ions that is not reflected in our final calculated selectivities. Entropic effects also contribute additional uncertainty to the calculations because it is difficult to predict the change in volume or rigidity of the surrounding polymer environment upon ion binding. Because of this, our free energy computations include only ligand translational and rotational entropy contributions (treating both binding partners as rigid bodies). Relative to Cl$^-$ (which has no rotational entropy), the entropy contributions decrease the overall affinity for all ions by 2.5–6 kcalmol$^{-1}$ (Supplementary Table 4).

The chemical portion of the binding free energies is provided by energy differences in minimum energy configurations. The energy difference itself favors binding of NO$_3^-$ and HSO$_4^-$ over TcO$_4^-$ by 7, 5, and 4 kcalmol$^{-1}$ for a, b, and d, respectively. This is not surprising, given that partial charges on NO$_3^-$ are most negative, followed by HSO$_4^-$ over TcO$_4^-$ in that order. When energetic and free energy differences are added together, Cl$^-$ is still the most favored binding partner in gas phase. However, the relatively hydrophobic environment of the polymer combined with the stronger affinity of water for Cl$^-$ results in the opposite trend of binding free energies in solution. Just as seen in experiments, the final order of ion exchange affinity in solution is TcO$_4^- >$ HSO$_4^- >$ NO$_3^- >$ Cl$^-$ for all polymers. d has the strongest relative affinity for TcO$_4^-$, at -6.1 kcal mol$^{-1}$ relative to Cl$^-$, while its affinity for HSO$_4^-$ is -3.9 and for NO$_3^-$ is $-3.8$ kcalmol$^{-1}$, which is in accordance with our experiment results (Supplementary Table 4).

**Stability test**. Considering the superior performance of PQA-$p$N(Me)$_2$Py-Cl to extract ReO$_4^-$ with good affinity and selectivity in the aforementioned situations, it was therefore the sample of choice for more detailed studies. The ability to regenerate and recycle the material would afford great advantages in reducing the overall cost and hence facilitating practical applications. To this end, the possibility of regeneration and reusing of PQA-$p$N(Me)$_2$Py-Cl were evaluated. Significantly, the ion exchange processes were fully reversible as reflected by the fact that the loaded ReO$_4^-$ ions can be washed off by saturated NaCl aqueous solutions with maintained ion-exchange capacity for at least three cycles, affording 990, 1003, and 987 mg g$^{-1}$, respectively (Fig. 5a). These results suggest the adequate chemical and structural stability of PQA-$p$N(Me)$_2$Py-Cl, which was further established by the following experiments. The ReO$_4^-$ uptake capacities and the pore structure of PQA-$p$N(Me)$_2$Py-Cl remained almost unchanged after being soaked in 12 M HCl or 2 M NaOH in saturated NaCl aqueous solution for one week (Fig. 5b and Supplementary Fig. 24). More impressively, PQA-$p$N(Me)$_2$Py-Cl was able to maintain its performance in extraction of ReO$_4^-$ from 3 M HNO$_3$ solution for more than three times with approximately 70% of ReO$_4^-$ from the HNO$_3$ solution containing 1.6 mM of ReO$_4^-$ (NO$_3^-$: ReO$_4^-$ molar ratio = 1875) at a phase ratio of 50 mLg$^{-1}$ after a single treatment (Fig. 5c). Notably, a removal efficiency of 97% can be reached at a phase ratio of 20 mLg$^{-1}$ (a number that is nearly half that used for the chromatographic column application, 11 mLg$^{-1}$). These features clearly represent a priority in

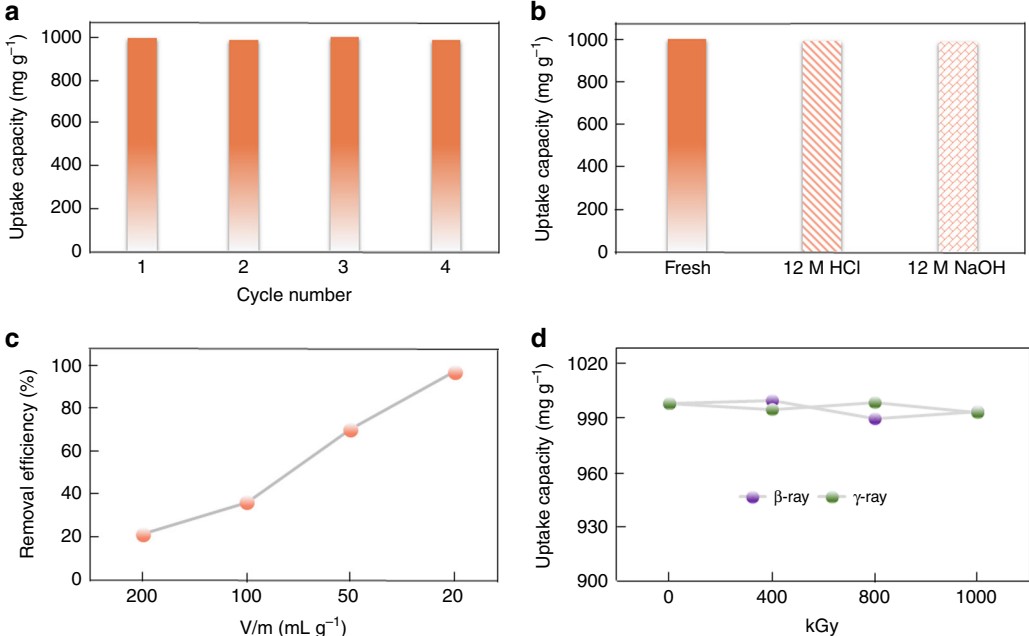

**Fig. 5** Stability evaluation. **a** Reversibility of PQA-*p*N(Me)$_2$Py-Cl for removing ReO$_4^-$. **b** ReO$_4^-$ uptake capacities of PQA-*p*N(Me)$_2$Py-Cl after being soaked in 12 M HCl or 2 M NaOH in saturated NaCl aqueous solution for one week. **c** Removal of ReO$_4^-$ by PQA-*p*N(Me)$_2$Py-Cl as a function of phase ratio (V/m) in 3 M HNO$_3$ aqueous solution. **d** Sorption capacities of ReO$_4^-$ by PQA-*p*N(Me)$_2$Py-Cl after being irradiated with varied doses of β-rays or γ-rays. Source data are provided as a Source Data file

uptake for ReO$_4^-$/TcO$_4^-$ over NO$_3^-$, thereby showing its potential for removing TcO$_4^-$ from reprocessed used fuel prior to the plutonium uranium redox extraction (PUREX) process.

Given that the above studies were done with non-radioactive elements, how radiation interacts with the materials was not taken into account[54]. Therefore, to rationalize results obtained from simulants, external radiation sources were exerted. Very impressively, PQA-*p*N(Me)$_2$Py-Cl exhibits outstanding resistance toward high-energy ionizing radiation of both β- and γ-rays, indicative by no noticeable change in the IR spectra, morphology, surface areas, ae well as the sorption performance measurements on the irradiated samples in relation to the pristine material, thereby providing a substantial prerequisite for application in used nuclear fuel reprocessing and waste management (Supplementary Figs. 25–27). Remarkably, even after irradiation with extremely large doses of 1000 kGy of β- or γ-rays, the sorption rate and capacity for ReO$_4^-$ by PQA-*p*N(Me)$_2$Py-Cl remained unaffected (Fig. 5d). PQA-*p*N(Me)$_2$Py-Cl combines the attributes of high capacity, reusability, and resistance against radiation, providing an extended lifetime that is sought after for long-term use in nuclear waste water treatment applications.

**Removal from simulated nuclear wastes**. To further validate the results, TcO$_4^-$ uptake experiments were performed by contacting PQA-*p*N(Me)$_2$Py-Cl with an aqueous solution of TcO$_4^-$. The concentration of TcO$_4^-$ was monitored by liquid scintillation counting measurements. As shown in Supplementary Fig. 28, PQA-*p*N(Me)$_2$Py-Cl was kinetically favored, 99% of TcO$_4^-$ could be removed within 5 min

With this success, it promoted us to investigate performance of PQA-*p*N(Me)$_2$Py-Cl in legacy nuclear waste. Previous estimates of Hanford tank waste inventories indicated that the amount of technetium in the waste needed to be reduced substantially to prepare immobilized low activity waste (ILAW) glass that meets performance assessment requirements. To investigate the removal of TcO$_4^-$ for this application, we prepared a simulated Hanford

LAW melter recycle stream, where the amounts of NO$_3^-$, NO$_2^-$, and Cl$^-$ are more than 300 times that of TcO$_4^-$, as listed in Supplementary Table 5, resulting in an enormous challenge to selectively remove TcO$_4^-$. Following similar experimental conditions as the established procedures (see details in the Supplementary Information), PQA-*p*N(Me)$_2$Py-Cl was found to remove about 95% of available TcO$_4^-$ from the waste at a phase ratio of 200 mL g$^{-1}$, placing it within striking distance of the all-time TcO$_4^-$ removal record (Supplementary Table 6). To further extend the potential application of PQA-*p*N(Me)$_2$Py-Cl in nuclear waste remediation, we tested its capability for the decontamination of TcO$_4^-$ in Savannah River Sites. It was shown to be effective for the segregation of TcO$_4^-$ from simplified simulants, a strong basic TcO$_4^-$ solution with the strength of other anions over 70 000 times that of TcO$_4^-$ (Supplementary Table 7). Approximately 80% of available TcO$_4^-$ was extracted from the waste at a phase ratio of 100 mL g$^{-1}$, which to date, we have found no materials in the literature with comparable selectivity under similar conditions. Although applicable to pertechnetate-containing waste streams in general, these two examples discussed here were directed at two specific potential applications at the U.S. Department of Energy's legacy site: i) the removal of dilute pertechnetate from near-neutral solutions, typical of the eluent streams from pretreated Hanford tank wastes, and ii) the direct removal of pertechnetate from highly alkaline solutions, typical of those found in Savannah River Sites.

## Discussion

In summary, based on the results above, we can now formulate clear design rules to achieve ideal ReO$_4^-$/TcO$_4^-$ scavengers with optimal uptake capacity and selectivity. The presented case shows that a subtle interplay between various functions in the adsorbent-in this case incorporation of a distinct functionality on a cation moiety-would be used to obtain an optimally performing adsorbent. Given the fact that ion exchange often follows a similar

mechanism, we expect that the trends discovered here will also be of high relevance for many other sequestration processes, such as coordinative binding. We conceptually showed the potential for tuning binding sites in selective capture to a greater degree of complexity. Given the demonstrated easy tunability, regeneration, and separation combined with radioactive resistance, long-term stability, and high performance in application relevant conditions, these new materials presented here could become highly influential for nuclear wastewater treatment technologies, particularly in the event of an impending water crisis. Also, potential cost savings and schedule acceleration could be anticipated based on consequent improvements in vitrification processing, reduction in waste form volumes, and higher waste-package performance.

## Methods

**Materials and measurements.** Commercially available reagents were purchased in high purity and used without purification. The purity and structure of the compounds synthesized in this manuscript were determined by NMR technique (Supplementary Fig. 29). Nitrogen sorption isotherms at the temperature of liquid nitrogen were measured using Micromeritics ASAP 2020M and Tristar system. $^1$H NMR spectra were recorded on a Bruker Avance-400 (400 MHz) spectrometer. Chemical shifts are expressed in ppm downfield from TMS at $\delta = 0$ ppm, and $J$ values are given in Hz. $^{13}$C (100.5 MHz) cross-polarization magic-angle spinning (CP-MAS) NMR experiments were recorded on a Varian infinity plus 400 spectrometer equipped with a magic-angle spin probe in a 4-mm ZrO$_2$ rotor. The samples were outgassed for 1000 min at 80 °C before the measurements. Scanning electron microscopy (SEM) and energy dispersive X-ray spectroscopy (EDX) mapping were performed on a Hitachi SU 8000. Transmission electron microscope (TEM) image was performed using a Hitachi HT-7700 or JEM-2100F field emission electron microscope (JEOL, Japan) with an acceleration voltage of 110 kV. High-angle-annular-dark-field (HAADF) scanning, STEM imaging, and energy dispersive X-ray spectroscopy (EDX) mapping were carried out by Titan ChemiSTEM operated at 200 kV. XPS spectra were performed on a Thermo ESCALAB 250 with Al Kα irradiation at $\theta = 90°$ for X-ray sources, and the binding energies were calibrated using the C1s peak at 284.9 eV. IR spectra were recorded on a Nicolet Impact 410 FTIR spectrometer. The concentrations of TcO$_4^-$ in solution were measured using a UV–vis spectrometer (Varian Cary 6000i) by monitoring the characteristic absorption peak at 290 nm and the activity of $^{99}$TcO$_4^-$ was also analyzed by a liquid scintillation counting (LSC) system (Perkin Elmer Quantulus 1220). ICP-OES was performed on a Perkin-Elmer Elan DRC II Quadrupole. ICP-MS was performed on a Perkin-Elmer Elan DRC II Quadrupole Inductively Coupled Plasma Mass Spectrometer. Elemental analyses were performed via flask combustion followed by ion chromatography.

## Data availability

The authors declare that all the data supporting the findings of this study are available within the article (and Supplementary Information files), or available from the corresponding author on reasonable request. The source data underlying Figs. 2, 3, and 5 are provided as a Source Data file.

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

## Acknowledgements

We acknowledge the US National Science Foundation (CBET-1706025) and the University of South Florida for financial support of this work.

## Author contributions

Q.S. and S.M. conceived and designed the research. Q.S. and B.A. performed the synthesis. Q.S., L.Z., P.K.T., and B.A. carried out the adsorption tests. C.X., J.C., and S.W. conducted the radiation stability tests. D. R. contributed to DFT calculation. All authors participated in drafting the paper, and gave approval to the final version of the manuscript.

## Additional information

**Competing interests:** The authors declare no competing interests.

