## [Peer Review File · Nature Communications]

Reviewers' comments:

Reviewer #1 (Remarks to the Author):

Innovative solid-phase sorbent technologies are needed to extract radio nuclides from complex media for energy security and environmental remediation. Sorbent materials densely furnished with selective binding sites with both high separation performance and adequate stability in these extremely challenging conditions are highly sought after. In this submission, the authors reported an efficient protocol for accessing a series of polymerizable monomers with densely arrayed pyridine moiety and their utilization in the formation of the anion nanotraps for TcO₄⁻ sequestration. The resultant adsorbents show high performance for TcO₄⁻ removal in a wide range of conditions in terms of both adsorption capacity and removal efficiency. The developed porous frameworks were sufficiently characterized along with very detailed sorption performance evaluation. In general, I think it is an important contribution to the field of nuclear waste management and will stimulate new thinking for the adsorbent design. Therefore, this reviewer recommends its publication after considering the following minor comments.

1. The authors gave comprehensive comparisons for the synthesized materials in terms of ReO₄⁻ segregation and these results showed that the anion nanotrap constructed with a dimethylamino group in the para position relative to pyridinium displayed extraordinary affinity for ReO₄⁻ with a record-breaking uptake capacity and selectivity under extreme conditions of super acidity and basicity, as well as high concentrations of competitors. To further highlight the superior performance of this material in TcO₄⁻ capture, the performances of the other two materials in the legacy nuclear wastes treatment are suggested to be evaluated.
2. In this submission, the authors show an elegant example that POPs constitute the newest avenue for high-performance adsorbents, displaying particular innovation and promise as potential technologies in the field of selective sequestration. To provide a more comprehensive information about POPs to the readers who do not work on this field, their applications in other pollutants removal are suggested to be included. Along this line, some important related references are suggested to be cited such as *Energy Environ. Sci.* 2011, 4, 3991.

Reviewer #2 (Remarks to the Author):

This is an interesting manuscript dealing with the continuous quest of searching ways to design adsorbent materials for efficient removal of toxic and radioactive species to provide enhanced protection as well as to facilitate closing the nuclear fuel cycle. The authors have apparently succeeded in finding an effective approach in this respect and with excellent performances in TcO₄⁻ ions segregation, one of the most troublesome radionuclide, from a range of water samples including simulated nuclear wastes. It is shown that the interaction between TcO₄⁻ ions and the sorbents can be optimized by modifying the local environment of the direct contact site, substantially overcoming the long-term challenge of TcO₄⁻ capture under extreme conditions of super acidity and basicity, strong irradiation field, and high ionic strength. The ion exchange behaviors of TcO₄⁻ on these materials were properly studied, showing a promising of such materials for further practical applications. Therefore, I would suggest the acceptance after minor revisions addressing the following comments:

- (1) To support the hierarchical structure of the synthesized materials, the corresponding pore size distributions are suggested to be provided.
- (2) To show the stability of the material under high-energy ionizing radiation, SEM images of the material after being exposed to β - and γ -rays are suggested to be provided.
- (3) To support the importance of migration of radionuclides for the environment, the following published papers are suggested to be cited: *Angew. Chem. Int. Ed.* 2015, 54, 12733-12737 and *Adv. Mater.* 2018, 30, 1801991.

Reviewer #3 (Remarks to the Author):

[Note from the editor: Please see also the attached pdf]

Overall evaluation

This manuscript suggested effective adsorption materials for selective separation of technetium existing as pertechnetate, TcO_4^- , in presence of many other oxyanion species. The materials were carefully synthesized and characterized by conventional techniques. Sorption experiment resulted in successful adsorption of ReO_4^- , which is an analogue of TcO_4^- . Density functional theory (DFT) analyses based on chemical binding point of view between ReO_4^- and the material indicated that thermochemical stability of ion-material complex in exchange reaction between each ion and Cl^- ion with respect to a material explains the order of the selectivity of oxyanions. Finally, the authors performed a recovery test of TcO_4^- ion from simulated nuclear waste.

I think that these are useful and applicable techniques for selective removal of TcO_4^- in the field of disposal of radioactive waste. The methodological and experimental details seem to be well-written. However, the discussion about a novelty of the materials is missing, although it is essential to strengthen the usability of the materials. The absence is not suitable for a publication of this manuscript in high-quality journal such as Nature Communications. Furthermore, computational details and results of DFT calculation were not enough to reproduce. Therefore, I recommend several revisions and clarifications of the following comments for the publication.

Introduction of this manuscript

Selective separation of technetate ion, TcO_4^- , has been required in the field of disposal of spent nuclear fuel and high-level radioactive liquid waste, which is generated during reprocessing of spent nuclear fuel, due to extremely long half-life and radiotoxicity of ^{99}Tc nuclide [1]. One of the difficulties in the selective separation is the presence of competitive oxyanion species, for examples, NO_3^- , SO_4^{2-} , and PO_4^{3-} , in the nuclear waste [2]. Authors of this manuscript have investigated the selective separation of $\text{TcO}_4^-/\text{ReO}_4^-$ using inorganic materials [3], resin [4-5], metal organic framework (MOF) [4-5], and porous organic polymer (POP) [6]. This study is classified into the development of POP materials for TcO_4^- removal.

The POP materials have porous organic polymeric structures with substituted methyl-pyridinium chloride or iodide moieties linked by methylene chain (Table 1). The authors synthesized their compounds by radical polymerization of vinyl pyridines, in which different three substituted groups are already introduced, followed by methylation of pyridine nitrogen with chlorine ion as counter (Supplementary Methods). They were characterized by NMR (Supplementary Figures 3, 7, 11, and 28), elementary analysis (Supplementary Table 3), XPS (Supplementary Figures 4, 8, 12, and 15-17), and electron microscopic techniques (Supplementary Figures 1-2, 6, 10, and 14). Procedures of $\text{TcO}_4^-/\text{ReO}_4^-$ sorption experiments and DFT analyses were described in "Supplementary Sorption Experiments and Computational Methods." Adsorption structure of ReO_4^- to the materials were characterized by SEM-EDX (Supplementary Figure 20), $\text{Re}(4f)$ XPS (Supplementary Figure 21), and infrared spectroscopy (Supplementary Figure 22). Recyclability, radiation-resistance, and TcO_4^- removal experiments were also tested by using the most effective candidate, PQA-*p*-N(Me) $_2$ Py-Cl.

Sorption experiments of ReO_4^- indicated that high extraction capacities were obtained for all materials. The performance, adsorption isotherms and kinetics,

increased in the order of PQA-Py-Cl, PQA-*p*-NH₂Py-Cl, and PQA-*p*-N(Me)₂Py-Cl. Selectivity of ReO₄⁻ over SO₄²⁻ or NO₃⁻ appeared for all the materials. The performance also increased in the same order to the extraction performance (Figure 2). Especially, the material, PQA-*p*-N(Me)₂Py-Cl, showed the highest selectivity of ReO₄⁻ over NO₃⁻ and SO₄²⁻ with 90 % removal towards 1000 times of concentration (Figure 3). DFT analyses indicated that, as described in eq. 1, Gibbs free energy difference, including solvent effect, in cation exchange reaction between Cl⁻ and X⁻ (= TcO₄⁻, HSO₄⁻, NO₃⁻) of simplified monomer model of the materials, M⁺, in which polymeric methylene groups were replaced into methyl groups and counter choline anion was omitted, was obtained as -6.1, -3.9, and -3.8 kcal mol⁻¹ for X = TcO₄⁻, HSO₄⁻, and NO₃⁻, respectively, in the case of simplified [PQA-*p*-N(Me)₂Py]⁺. This explained well the selectivity of TcO₄⁻ over SO₄²⁻ or NO₃⁻. High resistance to radiation (Figure 5) and recovery yield of TcO₄⁻ from simulated nuclear waste (Supplementary Figure 27) were also obtained.

Merit and criticism

The materials presented in this work were found to have very high extraction capacities and selectivity for technetate ion. Especially, the material, PQA-*p*-N(Me)₂Py-Cl, showed the best capacity for all the materials developed previously and applicative possibility to real system of nuclear waste management. The order of the performance in the selectivity of ReO₄⁻ was also explained theoretically by using quantum chemical calculations. I think that these points are newer compared to the authors' previous works and acceptable for publication in Nature Communications. However, two essential points (and minor points) needed to be addressed.

One is what is a novelty of this material? I see that this study is a sequel to the earlier study. The authors' previous study has already developed POP materials for TcO₄⁻ separation [6]. I cannot understand which point is newer. If the best performance is a newer point, the authors should note why the materials showed the best compared to previous materials. Second is what is a dominant in the cation-anion interaction? The authors varied parasite of the pyridinium moiety from -H to -NH₂ or -NMe₂ groups for POP material. I understood that this aims to adjust the geometric structural effect, including porous size of POP and steric hindrance of pyridinium moiety, and/or the electronic structural effect (chemical bonding). Please clarify these points.

As minor points, I also felt that the DFT computational details was not given in the present manuscript and supplementary information to reproduce. And I show unclear points I felt strange in the DFT calculations, as follows:

- i) I recommend that the authors submit the information as reader can reproduce this work. For examples, grid number and energy threshold value for self-consistent field calculation, Cartesian coordinates of simplified models, each Gibbs free energy values, evidence for usability of exchange-correlation functional should be described at least.
- ii) You showed the numerical data of Gibbs free energy difference in Supplementary Table 4, ΔG^0 , Solvation Free Energy, $\Delta\Delta G^{\text{solv}}$, ΔG^{exch} , and $\Delta\Delta G^{\text{exch}}$ (Relative to Cl⁻). However, the definitions and formulations of these values were not described, although the simple explanation was given in the mottom of the table. The authors must write the descriptions to avoid a misleading by the readers.

- iii) Why did you compare experimental ReO_4^- with calculated TcO_4^- , not ReO_4^- ? I understood that ReO_4^- is an analogue of TcO_4^- , however, I felt strange that it was not shown although it can be calculated.
- iv) Why did you compare experimental SO_4^{2-} with calculated HSO_4^- , not SO_4^{2-} ? Solvent effect, such as RISM, should compensate the difference in valence state. And have you tried the other oxyanions?
- v) Was there local minimum structure in the other conformation or geometrical configuration toward the materials of oxyanion? Please check and discuss about this possibility.

References

- [1] Ozawa, M.; Suzuki, S.; Takeshita, K. *Solv. Extr. Res. Dev. Jpn.*, **17**, 19-34 (2010).
- [2] USERDA, Alternatives for Managing Wastes from Reactors and Post-Fission Operations in the LWR Fuel Cycle, Vol. 2 Alternatives for Waste Treatment, ERDA-76-43, Sect. 6, pp.6_1-6_99 (1976).
- [3] Wang, S.; Yu, P.; Purse, B. A.; Orta, M. J.; Diwu, J.; Casay, W. H.; Phillips, B. L.; Alekseev, E. E.; Depmeier, W.; Hobbs, D. T.; Albrecht-Schmitt, T. E. *Adv. Funct. Mater.*, **22**, 2241-2250 (2012).
- [4] Zhu, L.; Xiao, C.; Dai, X.; Li, J.; Gui, D.; Sheng, D.; Chen, L.; Zhou, R.; Chai, Z.; Albrecht-Schmitt, T. E.; Wang, S. *Environ. Sci. Technol. Lett.*, **4**, 316-322 (2017).
- [5] Sheng, D.; Zhu, L.; Xu, C.; Xiao, C.; Wang, Y.; Wang, Y.; Chen, L.; Diwu, J.; Chen, J.; Chai, Z.; Albrecht-Schmitt, T. E.; Wang, S. *Environ. Sci. Technol.*, **51**, 3471-3479 (2017).
- [6] Li, J.; Dai, X.; Zhu, L.; Xu, C.; Zhang, D.; Silver, M. A.; Li, P.; Chen, L.; Li, Y.; Zuo, D.; Zhang, H.; Xiao, C.; Chen, J.; Diwu, J.; Farha, O. K.; Albrecht-Schmitt, T. E.; Chai, Z.; Wang, S. *Nat. Commun.*, **9**, 3007 (2018).

Date: February 20, 2019

We greatly appreciate the positive comments and constructive suggestions from the reviewers. We have revised the manuscript accordingly as detailed in the responses below, and the corresponding changes have been highlighted in yellow in the manuscript and supplementary information.

Reviewer #1:

Comment 1: Innovative solid-phase sorbent technologies are needed to extract radio nuclides from complex media for energy security and environmental remediation. Sorbent materials densely furnished with selective binding sites with both high separation performance and adequate stability in these extremely challenging conditions are highly sought after. In this submission, the authors reported an efficient protocol for accessing a series of polymerizable monomers with densely arrayed pyridine moiety and their utilization in the formation of the anion nanotraps for TcO_4^- sequestration. The resultant adsorbents show high performance for TcO_4^- removal in a wide range of conditions in terms of both adsorption capacity and removal efficiency. The developed porous frameworks were sufficiently characterized along with very detailed sorption performance evaluation. In general, I think it is an important contribution to the field of nuclear waste management and will stimulate new thinking for the adsorbent design. Therefore, this reviewer recommends its publication after considering the following minor comments.

We appreciate the reviewer's high comments and support of our work.

Comment 2: The authors gave comprehensive comparisons for the synthesized materials in terms of ReO_4^- segregation and these results showed that the anion nanotrap constructed with a dimethylamino group in the para position relative to pyridinium displayed extraordinary affinity for ReO_4^- with a record-breaking uptake capacity and selectivity under extreme conditions of super acidity and basicity, as well as high concentrations of competitors. To further highlight the superior performance of this material in TcO_4^- capture, the performances of the other two materials in the legacy nuclear wastes treatment are suggested to be evaluated.

We thank the reviewer for the valuable comments. Per the reviewer's suggestion, the TcO_4^- removal efficiency from the legacy nuclear wastes by the other two materials was evaluated and the corresponding data were included in Supplementary Tables 6 and 8.

Comment 3: In this submission, the authors show an elegant example that POPs constitute the newest avenue for high-performance adsorbents, displaying particular innovation and promise as potential technologies in the field of selective sequestration. To provide a more comprehensive information about POPs to the readers who do not work on this field, their applications in other

pollutants removal are suggested to be included. Along this line, some important related references are suggested to be cited such as Energy Environ. Sci. 2011, 4, 3991.

We thank the reviewer for the comments. A brief introduction of POPs was included and the related references were properly cited.

Reviewer #2:

Comment 1: This is an interesting manuscript dealing with the continuous quest of searching ways to design adsorbent materials for efficient removal of toxic and radioactive species to provide enhanced protection as well as to facilitate closing the nuclear fuel cycle. The authors have apparently succeeded in finding an effective approach in this respect and with excellent performances in TcO_4^- ions segregation, one of the most troublesome radionuclide, from a range of water samples including simulated nuclear wastes. It is shown that the interaction between TcO_4^- ions and the sorbents can be optimized by modifying the local environment of the direct contact site, substantially overcoming the long-term challenge of TcO_4^- capture under extreme conditions of super acidity and basicity, strong irradiation field, and high ionic strength. The ion exchange behaviors of TcO_4^- on these materials were properly studied, showing a promising of such materials for further practical applications. Therefore, I would suggest the acceptance after minor revisions addressing the following comments:

We appreciate the reviewer's valuable comments and support of our work.

Comment 2: To support the hierarchical structure of the synthesized materials, the corresponding pore size distributions are suggested to be provided.

We thank the reviewer for the comment. The materials' pore size distributions were included in the revised Supplementary Information.

Comment 3: To show the stability of the material under high-energy ionizing radiation, SEM images of the material after being exposed to β - and γ -rays are suggested to be provided.

We appreciate the reviewer for the valuable suggestion. The SEM images of the material after being exposed to the radioactive rays have been provided.

Comment 4: To support the importance of migration of radionuclides for the environment, the following published papers are suggested to be cited: Angew. Chem. Int. Ed. 2015, 54, 12733-12737 and Adv. Mater. 2018, 30, 1801991.

We thank the reviewer for the comment. These references have been properly cited.

Reviewer #3:

Comment 1:

Overall evaluation

This manuscript suggested effective adsorption materials for selective separation of technetium existing as pertechnetate, TcO_4^- , in presence of many other oxyanion species. The materials were carefully synthesized and characterized by conventional techniques. Sorption experiment resulted in successful adsorption of ReO_4^- , which is an analogue of TcO_4^- . Density functional theory (DFT) analyses based on chemical binding point of view between ReO_4^- and the material indicated that thermochemical

stability of ion-material complex in exchange reaction between each ion and Cl^- ion with respect to a material explains the order of the selectivity of oxyanions. Finally, the authors performed a recovery test of TcO_4^- ion from simulated nuclear waste I think that these are useful and applicable techniques for selective removal of TcO_4^- in the field of disposal of radioactive waste. The methodological and experimental details seem to be well-written. However, the discussion about a novelty of the materials is missing, although it is essential to strengthen the usability of the materials. The absence is not suitable for a publication of this manuscript in high-quality journal such as Nature Communications. Furthermore, computational details and results of DFT calculation were not enough to reproduce. Therefore, I recommend several revisions and clarifications of the following comments for the publication.

We thank the reviewer for the constructive comments and support of our work. The concerns raised by the reviewer have been responded point-by-point as listed below.

Comment 2:

Introduction of this manuscript

Selective separation of technetate ion, TcO_4^- , has been required in the field of disposal of spent nuclear fuel and high-level radioactive liquid waste, which is generated during reprocessing of spent nuclear fuel, due to extremely long half-life and radiotoxicity of ^{99}Tc nuclide [1]. One of the difficulties in the selective separation is the presence of competitive oxyanion species, for examples, NO_3^- , SO_4^{2-} , and PO_4^{3-} , in the nuclear waste [2]. Authors of this manuscript have investigated the selective separation of $\text{TcO}_4^-/\text{ReO}_4^-$ using inorganic materials [3], resin [4-5], metal organic framework (MOF) [4-5], and porous organic polymer (POP) [6]. This study is classified into the development of POP materials for TcO_4^- removal. The POP materials have porous organic polymeric structures with substituted methylpyridinium chloride or iodide moieties linked by methylene chain (Table 1). The authors synthesized their compounds by radical polymerization of vinyl pyridines, in which different three substituted groups are already introduced, followed by methylation of pyridine nitrogen with chlorine ion as counter (Supplementary Methods). They were characterized by NMR (Supplementary Figures 3, 7, 11, and 28), elementary analysis (Supplementary Table 3), XPS (Supplementary Figures 4, 8, 12, and 15-17), and electron microscopic techniques (Supplementary Figures 1-2, 6, 10, and 14). Procedures of $\text{TcO}_4^-/\text{ReO}_4^-$ sorption experiments and DFT analyses were described in "Supplementary Sorption Experiments and Computational Methods." Adsorption structure of ReO_4^- to the materials were characterized by SEM-EDX (Supplementary Figure 20), Re(4f) XPS (Supplementary Figure 21), and infrared spectroscopy (Supplementary Figure 22). Recyclability, radiation-resistance, and TcO_4^- removal experiments were also tested by using the most effective candidate, PQA- $p\text{N}(\text{Me})_2\text{Py}\text{-Cl}$. Sorption experiments of ReO_4^- indicated that high extraction capacities were obtained for all materials. The performance, adsorption isotherms and kinetics, increased in the order of PQA-Py-Cl, PQA- $p\text{NH}_2\text{Py}\text{-Cl}$, and PQA- $p\text{N}(\text{Me})_2\text{Py}\text{-Cl}$. Selectivity of ReO_4^- over SO_4^{2-} or NO_3^- appeared for all the materials. The performance also increased in the same order to the extraction performance (Figure 2). Especially, the material, PQA- $p\text{N}(\text{Me})_2\text{Py}\text{-Cl}$, showed the highest selectivity of ReO_4^- over NO_3^- and SO_4^{2-} with 90 % removal towards 1000 times of concentration (Figure 3). DFT analyses indicated that, as described in eq. 1, Gibbs free energy difference, including solvent effect, in cation exchange reaction between Cl^- and X^- (= TcO_4^- , HSO_4^- , NO_3^-) of simplified monomer model of the materials, M^+ , in which polymeric methylene groups were replaced into methyl groups and counter chroline anion was omitted, was obtained as -6.1, -3.9, and -3.8 kcal mol $^{-1}$ for $\text{X} = \text{TcO}_4^-$, HSO_4^- , and NO_3^- , respectively, in the case of simplified $[\text{PQA-}p\text{N}(\text{Me})_2\text{Py}]^+$. This explained well the selectivity of TcO_4^- over SO_4^{2-} or NO_3^- . High resistance to radiation

(Figure 5) and recovery yield of TcO_4^- from simulated nuclear waste (Supplementary Figure 27) were also obtained.

We appreciate the reviewer for taking the time to review our work.

Comment 3:

Merit and criticism

The materials presented in this work were found to have very high extraction capacities and selectivity for technetate ion. Especially, the material, PQA- $p\text{N}(\text{Me})_2\text{Py}-\text{Cl}$, showed the best capacity for all the materials developed previously and applicative possibility to real system of nuclear waste management. The order of the performance in the selectivity of ReO_4^- was also explained theoretically by using quantum chemical calculations. I think that these points are newer compared to the authors' previous works and acceptable for publication in Nature Communications. However, two essential points (and minor points) needed to be addressed. One is what is a novelty of this material? I see that this study is a sequel to the earlier study. The authors' previous study has already developed POP materials for TcO_4^- separation [6]. I cannot understand which point is newer. If the best performance is a newer point, the authors should note why the materials showed the best compared to previous materials. Second is what is a dominant in the cation-anion interaction? The authors varied para-site of the pyridinium moiety from -H to - NH_2 or - NMe_2 groups for POP material. I understood that this aims to adjust the geometric structural effect, including porous size of POP and steric hindrance of pyridinium moiety, and/or the electronic structural effect (chemical bonding). Please clarify these points.

We thank the reviewer for the criticisms. Seemingly, there are parallels between the previous Nature Communication paper -- that advanced our understanding and the challenges to sequester TcO_4^- -- and the current submission. Our new work has made great strides from the earlier research, whereby we have introduced multiple functionalities in the adsorbents, rather than relying on a single direct interaction. We describe that the interaction with TcO_4^- can be altered by modifying the local environment of the cationic moieties, as demonstrated by manipulating the affinity of pyridinium-based anion nanotraps toward pertechnetate. Additionally, we have redesigned the entire material to improve its properties and to better elucidate the structure-property relationships while also thinking about its future application on an industrial scale. Specifically, these two materials are both classified as organic polymers, yet they have fundamental differences in their synthetic procedures and chemical compositions. The previously reported cationic polymeric network was synthesized by the condensation between 1,1,2,2-tetrakis(4-(imidazolyl-4-yl)phenyl)ethene and 1,4-bis(bromomethyl)benzene. The resulting material is nonporous, unfavorable for adsorption kinetics, which largely defines the adequate flow rate for decontamination of polluted water and the subsequent efficiency of the process. In addition, the sensitivity of the imidazole ring toward strong basic solutions limits its application. Moreover, this material is notoriously expensive due to the costly reagents. In our current submission, the anion nanotraps were constructed by the functional pyridinium moieties alone, to conserve a sufficiently high density of accessible exchange sites and also to ensure the composition for facilitating the establishment of structure-property relationships. Furthermore, the free radical polymerization method employed is simpler and more cost-effective, a must for any future industrial-scale processes, and also with great monomer tolerance, beneficial for any performance modification. With regard to the results, the newly developed adsorbents outperforms the

previous one in nearly all aspects. Given the greatly improved stability toward strong basic aqueous solutions, the newly synthesized anion nanotraps can be used to selectively sequester the TcO_4^- anions from Savannah River Sites. In summary, systematic control of the substituent effect allows the resulting anion nanotraps to combine multiple features as ideal pertechnetate scavengers with exceptional performances, substantially overcoming the long-term challenge of TcO_4^- segregation under extreme conditions of super acidity and basicity, strong irradiation field, and high ionic strength. We conceptually showed the potential for tuning binding sites in selective capture to a greater degree of complexity.

It was found that under all conditions tested, the preference of these pyridinium-based sorbents towards ReO_4^- complexation follows the same order: $\text{PQA-}p\text{N}(\text{Me})_2\text{Py-Cl} > \text{PQA-Py-Cl} > \text{PQA-}p\text{NH}_2\text{Py-Cl}$. This trend may not be pinpointed to a single property change of the adsorbents but rather as a consequence of a complex interplay of several factors. For example, an increase in surface area may play a key role in the uptake capacity by providing a greater number of exposed exchangeable sites. However, in the present case, only a very weak correlation with the surface area of these materials could be established, thus suggesting that surface area is not the central factor determining the sorption performance; rather, the interaction strength between binding sites and guest species is a more important factor for determining guest uptake. Intriguingly, the increase in gravimetric capacity in going from $\text{PQA-}p\text{N}(\text{Me})_2\text{Py-Cl}$ to PQA-Py-Cl and $\text{PQA-}p\text{NH}_2\text{Py-Cl}$ is opposite to what would be expected from the density of ion-exchange sites. For instance, on the basis of elemental analysis, it was found to have a Cl^- species content of 10.3, 11.1, and 12.5 wt.%, for $\text{PQA-}p\text{N}(\text{Me})_2\text{Py-Cl}$, PQA-Py-Cl , and $\text{PQA-}p\text{NH}_2\text{Py-Cl}$, respectively. We therefore attribute this anomalous increase in ReO_4^- uptake capacity for $\text{PQA-}p\text{N}(\text{Me})_2\text{Py-Cl}$ to the higher electrostatic ion-pairing attraction energies between the cationic polymer backbone of $\text{PQA-}p\text{N}(\text{Me})_2\text{Py-Cl}$ and ReO_4^- ions, which was supported by the calculated and experimentally determined distribution coefficient values. We have discussed this at page 12 (Intriguingly, the increase in..... $\text{PQA-}p\text{N}(\text{Me})_2\text{Py-Cl}$ and ReO_4^- ions) and page 15 (Under all conditions testedfor determining guest uptake).

In summary, we took the initiative to build and improve upon the initial studies to design better materials with refined applications and consider their long term potential for a more economically feasible synthesis for large-scale production processes.

As minor points, I also felt that the DFT computational details was not given in the present manuscript and supplementary information to reproduce. And I show unclear points I felt strange in the DFT calculations, as follows:

We thank the reviewer for pointing these out and please see the point-by-point response as follows:

- i) I recommend that the authors submit the information as reader can reproduce this work. For examples, grid number and energy threshold value for self-consistent field calculation, Cartesian coordinates of simplified models, each Gibbs free energy values, evidence for usability of exchange-correlation functional should be described at least.
Grid spacing and energy tolerance information were added to the main text in the subsection "Density functional theory calculation studies" and to the Supplementary Information where the basis set is mentioned. We cite Supplementary Refs 1 and 2,

Williams 2014 and Weaver 2017 as evidence that the XC functional and basis set combinations are usable for our study. The citations have been added to the main text as well.

- ii) You showed the numerical data of Gibbs free energy difference in Supplementary Table 4, ΔG^0 , Solvation Free Energy, $\Delta\Delta G_{\text{solv}}$, ΔG_{exch} , and $\Delta\Delta G_{\text{exch}}$ (Relative to Cl^-). However, the definitions and formulations of these values were not described, although the simple explanation was given in the mottom of the table. The authors must write the descriptions to avoid a misleading by the readers.

Three new equations (S.5-S.7) have been added to provide explicit formulas for the free energies appearing in Supplementary Table S4.

- iii) Why did you compare experimental ReO_4^- with calculated TcO_4^- , not ReO_4^- ? I understood that ReO_4^- is an analogue of TcO_4^- , however, I felt strange that it was not shown although it can be calculated.

The main text now explains that our quantum methods have been tested for TcO_4^- but not for ReO_4^- .

- iv) Why did you compare experimental SO_4^{2-} with calculated HSO_4^- , not SO_4^{2-} ? Solvent effect, such as RISM, should compensate the difference in valence state. And have you tried the other oxyanions?

The main text on p. 17 (just before Fig. 4) explains that SO_4^{2-} is subject to a larger systematic numerical error than HSO_4^- , and that any differences in energies between the two can be understood in terms of pKa values in the complex vs. in solution. RISM should technically compensate for differences in the valence state, but may have larger error because of the approximations inherent in the method. More importantly, the gas-phase cluster approximation does not apply as well to SO_4^{2-} as to HSO_4^- . Both errors come from the presence of much more tightly bound waters for complexes with SO_4^{2-} . Our gas-phase QM calculations with SO_4^{2-} showed partially covalent attachment that would have been lessened by including explicit waters. However, the additional waters would have increased the complexity of the model.

- v) Was there local minimum structure in the other conformation or geometrical configuration toward the materials of oxyanion? Please check and discuss about this possibility.

The main text now states explicitly that all starting conformations initially oriented an oxygen of the anionic ligand toward the ring nitrogen. The conformational differences observed -- particularly where two oxyanions sit on either side of the ring nitrogen -- indicate an energetic preference for these configurations over the intuitive, directly aligned binding conformation.

References

- [1] Ozawa, M.; Suzuki, S.; Takeshita, K. *Solv. Extr. Res. Dev. Jpn.*, 17, 19-34 (2010).
[2] USERDA, Alternatives for Managing Wastes from Reactors and Post-Fission Operations in the LWR Fuel Cycle, Vol. 2 Alternatives for Waste Treatment, ERDA-76-43, Sect. 6, pp.6_1-6_99 (1976).

- [3] Wang, S.; Yu, P.; Purse, B. A.; Orta, M. J.; Diwu, J.; Casay, W. H.; Phillips, B. L.; Alekseev, E. E.; Depmeier, W.; Hobbs, D. T.; Albrecht-Schmitt, T. E. *Adv. Funct. Mater.*, 22, 2241-2250 (2012).
- [4] Zhu, L.; Xiao, C.; Dai, X.; Li, J.; Gui, D.; Sheng, D.; Chen, L.; Zhou, R.; Chai, Z.; Albrecht-Schmitt, T. E.; Wang, S. *Environ. Sci. Technol. Lett.*, 4, 316-322 (2017).
- [5] Sheng, D.; Zhu, L.; Xu, C.; Xiao, C.; Wang, Y.; Wang, Y.; Chen, L.; Diwu, J.; Chen, J.; Chai, Z.; Albrecht-Schmitt, T. E.; Wang, S. *Environ. Sci. Technol.*, 51, 3471-3479 (2017).
- [6] Li, J.; Dai, X.; Zhu, L.; Xu, C.; Zhang, D.; Silver, M. A.; Li, P.; Chen, L.; Li, Y.; Zuo, D.; Zhang, H.; Xiao, C.; Chen, J.; Diwu, J.; Farha, O. K.; Albrecht-Schmitt, T. E.; Chai, Z.; Wang, S. *Nat. Commun.*, 9, 3007 (2018).

Again we thank the reviewers for the constructive comments and suggestions, which have made our manuscript much improved.

REVIEWERS' COMMENTS:

Reviewer #1 (Remarks to the Author):

It is acceptable for publication.

Reviewer #2 (Remarks to the Author):

The authors have satisfactorily addressed all the comments from the reviewers and the work can be accepted as is.

Reviewer #3 (Remarks to the Author):

This revised manuscript and the authors' comments have addressed my questions and minor points. The authors developed new materials for a separation of pertechnetate by using fundamentally different methods toward previous studies, including chemical framework, radical polymerization, etc. They also achieved substantial improvements, including adsorption rate, tolerance and economic cost. I am also interested in "structure-property relationship" that the authors suggested in developing separation materials, because the performance in uptake capacity for this kind of materials can be correlated to electrostatic interaction between the pyridinium moiety and the metal ion. I agree and support this strategy and this might lead to efficient design of materials. I checked that the minor points for DFT calculation part has been correctly revised. Finally, I can recommend this manuscript to be published in Nature Communications without any changes.

Reviewer #1

Comment: It is acceptable for publication.

We are grateful to the reviewer for taking the time to evaluate our work and support from the reviewer.

Reviewer #2

Comment: The authors have satisfactorily addressed all the comments from the reviewers and the work can be accepted as is.

We thank the reviewer for taking the time to evaluate our work and support from the reviewer.

Reviewer #3

Comment: This revised manuscript and the authors' comments have addressed my questions and minor points. The authors developed new materials for a separation of pertechnetate by using fundamentally different methods toward previous studies, including chemical framework, radical polymerization, etc. They also achieved substantial improvements, including adsorption rate, tolerance and economic cost. I am also interested in "structure-property relationship" that the authors suggested in developing separation materials, because the performance in uptake capacity for this kind of materials can be correlated to electrostatic interaction between the pyridinium moiety and the metal ion. I agree and support this strategy and this might lead to efficient design of materials. I checked that the minor points for DFT calculation part has been correctly revised. Finally, I can recommend this manuscript to be published in Nature Communications without any changes.

We appreciate the reviewer's high comments and support of our work.